# Effect of Cyclic Loading on Mode I Fracture Toughness of Granite under Real-Time High-Temperature Conditions

**Fei Lv, Fan Zhang \*, Subiao Zhang, Kangwen Li and Shuangze Ma**

School of Civil Engineering, Architecture and Environment, Hubei University of Technology, Wuhan 430068, China; 102000707@hbut.edu.cn (F.L.); 102110855@hbut.edu.cn (S.Z.); 102110886@hbut.edu.cn (K.L.); 102110853@hbut.edu.cn (S.M.)

\* Correspondence: fanzhang@hbut.edu.cn

**Abstract:** Under hot dry rock development, rock formations undergo the combined challenges of cyclic loading and high temperatures, stemming from various sources such as cyclic hydraulic fracturing and mechanical excavation. Therefore, a fundamental understanding of how rocks fracture under these demanding conditions is fundamental for cyclic hydraulic fracturing technology. To this end, a series of three-point bending tests were conducted on granite samples. These tests entailed exposing the samples to cyclic loading under varying real-time high-temperature environments, ranging from 25 °C to 400 °C. Furthermore, different upper load limits (75%, 80%, 85%, and 90% of the peak load) obtained in monotonic three-point bending tests were used to explore the behavior of granite under these conditions. The analysis encompassed the study of load–displacement curves, elastic stiffness, and mode I fracture toughness under cyclic loading conditions. In addition, the microscopic features of the fracture surface were examined using a scanning electron microscope (SEM). The findings revealed notable patterns in the behavior of granite. Cumulative vertical displacement in granite increased with the growing number of cycles, especially at 25 °C, 200 °C, and 300 °C. This displacement exhibited a unique trend, initially decreasing before subsequently rising as the cycle count increased. Additionally, the critical damage threshold of granite exhibited a gradual decline as the temperature rose. As the temperature ascended from 25 °C to 200 °C, the damage threshold typically ranged between 80% and 85% of the peak load. At 300 °C, this threshold declined to approximately 75–80% of the peak load, and at 400 °C, it fell below 75% of the peak load. Within the temperature ranging from 25 °C to 300 °C, we noted a significant increase in the incidence of cracks, crystal microfracture zones, and the dislodging of mineral particles within the granite as the number of cycles increased.

**Keywords:** granite; high temperature; cyclic loading; fracture toughness; damage threshold

## 1. Introduction

Widely recognized as a clean and sustainable geological resource, hot dry rocks (HDRs) have garnered global interest as conventional fossil fuels are depleted, and their environmental consequences have become increasingly evident. The primary method for harnessing HDRs is the enhanced geothermal system (EGS) [1,2]. The EGS concept aims to establish highly permeable fractured zones to facilitate fluid flow and heat extraction through hydraulic stimulation [3–5]. In recent studies, researchers have demonstrated that conventional hydraulic fracturing is susceptible to inducing reservoir seismicity, an issue that has led to the discontinuation of specific geothermal projects [6,7].

As an alternative to traditional hydraulic fracturing, cyclic hydraulic fracturing has shown promise in effectively reducing both the crack initiation pressure and fracture toughness of HDRs, mitigating the risk of water injection-induced seismicity [8,9]. Compared to conventional hydraulic fracturing methods, cyclic hydraulic fracturing techniques offer distinct advantages by promoting gradual and controlled fracture growth. This phenomenon

may be attributed to the cyclic loading process, which induces rock softening, hence facilitating easier fracture formation and reducing the initiation pressure, as suggested by Patel et al. [10] and Zang et al. [11]. Alternatively, the cyclic hydraulic fracturing process enables the gradual release of energy stored within the reservoir, effectively lowering the seismic risk. Moreover, within the fatigue process, cracks exhibit increased flexibility in multiple orientations, promoting the formation of interconnected reticulated cracks and expanding the permeable areas, as observed in recent findings by Li et al. [12].

Nevertheless, the impact of cyclic hydraulic fracturing on fracture toughness in the context of HDR mining remains overlooked. Fracture toughness serves as a critical measure for assessing a rock's resistance to cracking and the expansion of fractures, as noted by Nasseri et al. [13]. To enhance the efficiency of heat mining in EGS projects, examining the alterations in fracture toughness and the macro/microstructure due to cyclic loading in three-point bending (CTB) tests conducted under high-temperature conditions is a fundamental task.

Numerous studies have investigated rocks' physical and mechanical properties following thermal treatment. For instance, Arifi et al. [14] observed that the mechanical properties of granite undergo significant alterations at elevated temperatures, often attributed to transformations in rock components and crystalline phases. In a similar context, Balme et al. [15] conducted high-temperature experiments on three distinct varieties of basalts, uncovering intriguing findings. Their results indicated that, at 150 °C, the fracture toughness of Icelandic basalts surpasses that of the other types examined. Moreover, research has reported a significant decline in the fracture toughness of brittle rocks as temperatures rise, particularly beyond the 400 °C threshold, as noted by Nasseri et al. [13]. Xue et al. [16] also analyzed fracture propagation characteristics during the water-cooling shock process and introduced a comprehensive thermal–hydraulic–mechanical coupling model, highlighting the factors contributing to rock fracture reduction. In another study, Alneasan et al. [17] discussed the impact of heat treatment on the fracture characteristics of mudstone, offering insights into the mechanism behind the observed increase in fracture toughness following heat treatment. Notably, several studies have also reported a rise in fracture toughness for brittle rocks within the temperature range of 25 °C to 100 °C, as demonstrated by Meredith et al. [18], Mahanta et al. [19], Zhang et al. [20], Feng et al. [21], and Zheng et al. [22]. This enhancement of fracture toughness is mainly attributed to the closure of microcracks and pores due to thermal expansion.

On the other hand, cyclic loading significantly affects the physical properties and mechanical response of granite. Recent findings indicated that the fatigue failure of rocks can be categorized into three distinct stages: the initial deformation stage, the constant deformation stage, and the accelerated deformation stage, as detailed in the studies of Wang et al. [23,24]. Extensive research has been conducted to investigate the impact of cyclic loading on the fracture toughness of rocks [25–29]. The results indicated that the fracture toughness of the specimens experiences a noticeable decrease when subjected to cyclic loading. Regarding cyclic loading frequency, Zhu et al. [30] and Ding et al. [31] examined the connection between cyclic loading frequency and crack propagation. Their findings demonstrated that at higher frequencies, such as 1 Hz, cracks exhibit rapid growth in terms of length, number, and width. In more recent research, Wang et al. [32] conducted a comprehensive analysis of the connection between dynamic fracturing, failure modes, and acoustic emission signals in marble subjected to freeze–thaw and stress cycles. As such, several factors are found to affect the rock fatigue mechanical property, including maximum stress, stress amplitude, and loading waveform, as demonstrated by Peng et al. [33] and Wang et al. [34,35].

The investigation into the fatigue mechanical behavior of granite at real-time temperatures remains relatively overlooked in the current literature. However, in the environment of cyclic hydraulic fracturing, HDRs experience a coupled environment characterized by high temperatures and cyclic loading. Consequently, this study presents a novel approach by conducting cyclic loading and unloading experiments under real-time temperature con-

ditions, aiming to simulate actual operational scenarios more effectively. In this regard, the present study focuses on the examination of fracture mechanisms and the alterations in the initial load of granite, with a particular emphasis on changes in mechanical properties such as cumulative vertical displacement (CVD), elastic stiffness, fracture toughness, and the reduction in fracture toughness under the combined effects of cyclic loading and real-time high temperatures. In recent years, there has been a growing demand for extracting HDR resources at depths shallower than 5 km, with temperatures ranging from 100 °C to 400 °C. This shift is driven by economic considerations and equipment limitations, as noted by Mccartney et al. [36]. To achieve this, we conducted three-point bending (CTB) tests on granite samples under various real-time high-temperature conditions and implemented different upper load limits, ranging from 75% to 90% of the peak load. We investigated the alterations in load–displacement curves, CVD, elastic stiffness, and fracture toughness, comparing them with results from monotonic three-point bending (MTB) tests. Subsequently, the damage morphology after CTB tests was observed using scanning electron microscopy (SEM). Lastly, we conducted an in-depth analysis of the influence of high temperature and cyclic loading on the crack initiation pressure in three-point bending tests. The findings of this research provide novel perspectives on the fatigue fracturing processes of granite within deep geothermal systems.

## 2. Materials and Methods

### 2.1. Materials

Granite samples were meticulously extracted from granite blocks sourced in Shuitou Town, located in China's Fujian Province. The blue-grey sample surfaces exhibited no defects. The granite possesses a natural density of 2.78 g/cm$^3$ and a porosity of 0.52%. Petro-physical analysis conducted with the X-ray diffraction technique revealed the mineral composition of the granite, predominantly comprising albite (39.63%), biotite (39.15%), quartz (16.76%), and tremolite (4.46%). Out of the four sample types recommended by the International Society of Rock Mechanics and Rock Engineering (ISRM) for measuring mode I fracture toughness [37–39], we chose the semi-circular bend (SCB) sample for its practical testing convenience, as highlighted by Bahrami et al. [40]. The granite sample's geometry is depicted in Figure 1, with its geometrical specifications conforming to the ISRM's recommended tolerances, as detailed in Table 1.

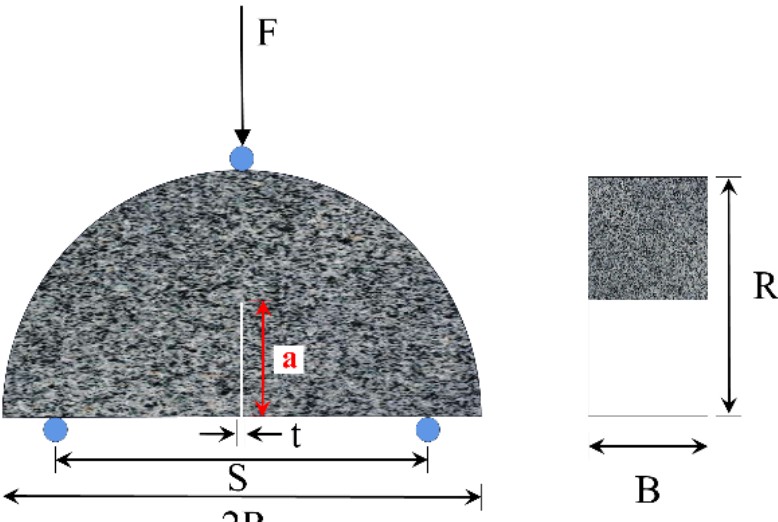

**Figure 1.** Geometry for the semicircular bend (SCB) sample, with R referring to the radius, B being the thickness, S being the distance between two supporting rollers, t being the thickness of the pre-crack, a being the length of the pre-crack, and F being the load.

**Table 1.** Basic geometrical parameters of granite samples and the three-point bending fixture, where R is the radius, B is the thickness, t is the thickness of the pre-crack, a is the length of the pre-crack, and S is the distance between two supporting rollers.

| R/(mm) | B/(mm) | t/(mm) | a/(mm) | S/(mm) |
|--------|--------|--------|--------|--------|
| $38 \pm 0.3$ | $30 \pm 0.2$ | 1 | $19.5 \pm 0.3$ | 60.8 |

*2.2. Methods*

The experiment used an integrated setup that included loading equipment, a heating apparatus, and a control and data acquisition system, enabling three-point bending tests under varying real-time temperature conditions. Figure 2 depicts the loading system comprising a reaction frame, fixture base, pressure bar, and fixture, with a maximum axial pressure capacity of 300 kN. The heating equipment consists of a temperature chamber and a heat exchange machine, encompassing a temperature range from −70 to 400 °C with an accuracy of ±0.1 °C, as illustrated in Figure 3. The temperature control screen on the heat exchange machine's side facilitates the adjustment of the temperature rise rate and the target temperature. The control and data acquisition system includes a force–displacement sensor and a terminal computer capable of collecting force–displacement data at 30 Hz. As a reference value in CTB tests, the peak load $P_{maxT}$ from the MTB test conducted at a loading rate of 0.1 mm/min was recorded and is presented in Table 2, following the approach presented by Andrea et al. [41].

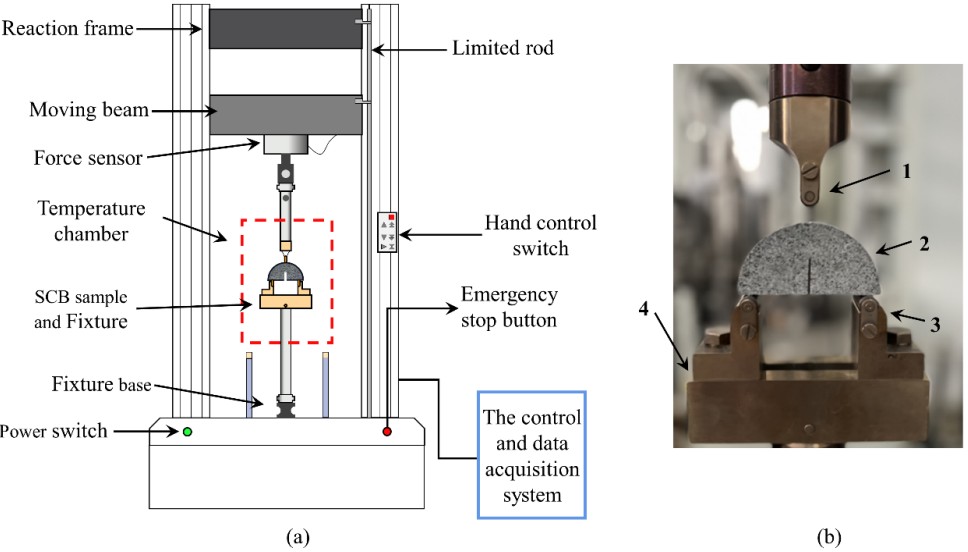

(a)

(b)

**Figure 2.** Configuration of the loading equipment: (**a**) testing machine and (**b**) three-point bending fixture with (1) compression roller, (2) SCB sample, (3) supporting roller, and (4) pedestal.

The SCB samples were categorized into 16 distinct groups. To maintain the accuracy and reliability of the test, each working condition was assessed with three separate samples. During the preliminary tests, it was observed that when the upper load limit exceeded 90%, the samples frequently failed before reaching the upper load limit. Conversely, when the upper load limit was set below 75%, it proved challenging for the samples to reach the point of failure. Therefore, the CTB tests were conducted with four different upper load limits (75%, 80%, 85%, and 90% of $P_{maxT}$) and at four different temperature levels (25 °C, 200 °C, 300 °C, and 400 °C), respectively. In the initial step, the granite sample was positioned at the center of the three-point bending fixture. Next, the sample was centered within the temperature chamber by pushing the chamber into position. Subsequently, the temperature chamber was securely fixed, and the chamber door was closed. Concurrently, insulation cotton was introduced into the gap between the connecting

rod and the temperature chamber to minimize heat loss and maintain a stable temperature environment. Throughout the heating process, temperature gradually increased to the specified levels (25 °C, 200 °C, 300 °C, and 400 °C) at a predefined rate of 10 °C per minute. This heating rate was adopted to minimize the impact of thermal gradients within the sample, following the approach outlined by Yin et al. [42]. Subsequently, the temperature chamber was maintained at the target temperature for 2 h to ensure uniform heating of the samples, in line with the methodology recommended by Zhang et al. [43,44]. After the temperature stabilization phase, the sample was subjected to CTB tests under the same loading and heating conditions as the in MTB test. This process continued until either the sample reached failure or the predefined limit of 200 loading cycles. In the CTB tests, it is essential to note that the lower load limit, represented by the lowest load value in each cycle, was consistently set at 150 N for all the samples. The cyclic loading path in the CTB tests is visually illustrated in Figure 4. These CTB tests followed a displacement-controlled approach with a 0.1 mm/min loading rate. After the mechanical tests, the fracture specimens were chosen for gold sputtering treatment on their fracture surfaces. The microscopic features of the fracture surface were examined using a scanning electron microscope at 500× magnification. The scanning electron microscope employed in this experiment offers a resolution spanning from 3 nm to 15 nm, a magnification range of ×5 to ×300,000, and an acceleration voltage range of 0.3 kV to 30 kV.

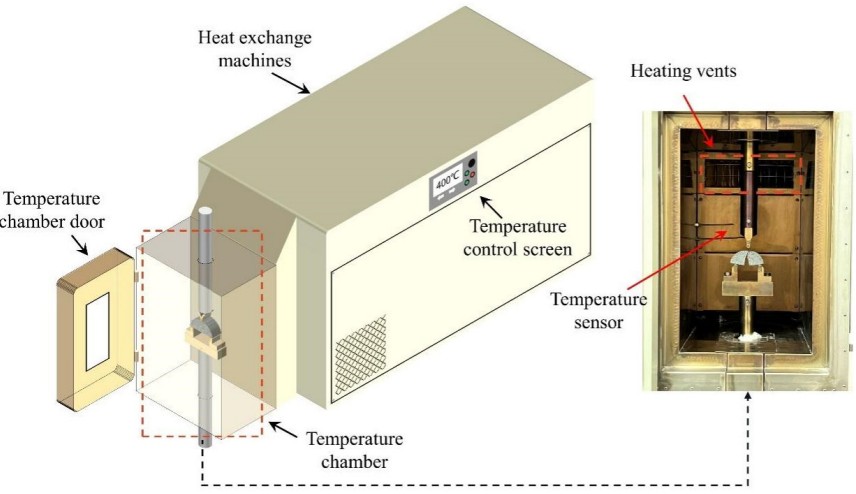

**Figure 3.** Heating equipment configuration.

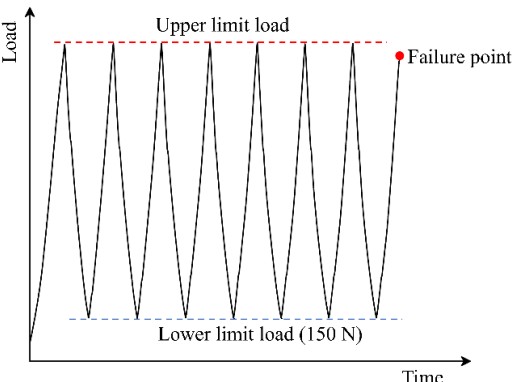

**Figure 4.** Schematic illustration of the load's dynamic evolution in cyclic loading, where the red dashed line represents the peak load during CTB, the blue dashed line indicates the lowest load point during CTB, and the red dot signifies the load at which sample failure occurs.

**Table 2.** Tested parameters of temperature and cyclic loading during CTB tests.

| Temperature (°C) | Peak Load of MTB Tests (N) | CTB Upper Load Limit | CTB Lower Limit Load (N) |
|---|---|---|---|
| 25 | $P_{max(25\,°C)}$ = 1992 | 75% $P_{max(25\,°C)}$<br>80% $P_{max(25\,°C)}$<br>85% $P_{max(25\,°C)}$<br>90% $P_{max(25\,°C)}$ | 150 |
| 200 | $P_{max(200\,°C)}$ = 1693 | 75% $P_{max(200\,°C)}$<br>80% $P_{max(200\,°C)}$<br>85% $P_{max(200\,°C)}$<br>90% $P_{max(200\,°C)}$ | 150 |
| 300 | $P_{max(300\,°C)}$ = 1481 | 75% $P_{max(300\,°C)}$<br>80% $P_{max(300\,°C)}$<br>85% $P_{max(300\,°C)}$<br>90% $P_{max(300\,°C)}$ | 150 |
| 400 | $P_{max(400\,°C)}$ = 1441 | 75% $P_{max(400\,°C)}$<br>80% $P_{max(400\,°C)}$<br>85% $P_{max(400\,°C)}$<br>90% $P_{max(400\,°C)}$ | 150 |

In this research, the mode I fracture toughness of granite was determined according to the ISRM-recommended formula, as noted by Kuruppu et al. [45]:

$$K_{Ic} = Y' \frac{P_{max}\sqrt{\pi a}}{2RB} \tag{1}$$

where $K_{Ic}$ denotes the mode I fracture toughness; $a$ is the length of the pre-crack; $P_{max}$ is the fracture load in the last cycle; $R$ and $B$ are the radius and thickness of the sample, respectively; and $Y'$ denotes the I non-dimensionless stress intensity factor, which can be expressed as follows:

$$Y' = -1.297 + 9.516\left(\frac{S}{2R}\right) - \left(0.47 + 16.457\left(\frac{S}{2R}\right)\right)\beta + \left(1.071 + 34.401\left(\frac{S}{2R}\right)\right)\beta^2 \tag{2}$$

where $S$ refers to the distance between the two supporting rollers: $\alpha = S/2R = 0.8$ and $\beta = L/R = 0.5$.

## 3. Results and Discussion

### 3.1. Load–Displacement Curve

Figure 5 illustrates the load–displacement curves for CTB tests conducted on granite at 400 °C, each with varying upper load limits. To differentiate between the fracture load of granite samples in MTB and CTB tests, it is essential to clarify that the peak load refers to the fracture load in MTB tests, while the damage load refers to the fracture load in CTB tests. Notably, a significant memory effect is observed in the granite's behavior during the cyclic loading. The load–displacement curves of the first cycle closely resemble those observed in MTB tests. Nonetheless, as the number of cycles increases, even when the upper load limit is below the peak load observed in MTB tests, the microcracks within the sample gradually connect, resulting in sample failure. Notably, at 400 °C, in comparison to the peak load in MTB tests, the damage load of samples subjected to CTB tests with upper load limits of 75%, 80%, 85%, and 90% experiences a reduction of 27.14%, 19.38%, 19.55%, and 13.36%, respectively, expressed as the percentage reduction of the CTB damage load to the peak load in MTB tests at the same temperature. Across different temperature settings. i.e., 25 °C, 200 °C, and 300 °C, the test outcomes consistently demonstrate similar patterns, with the damage load in CTB tests increasing as the upper load limit rises, as summarized in Table 3.

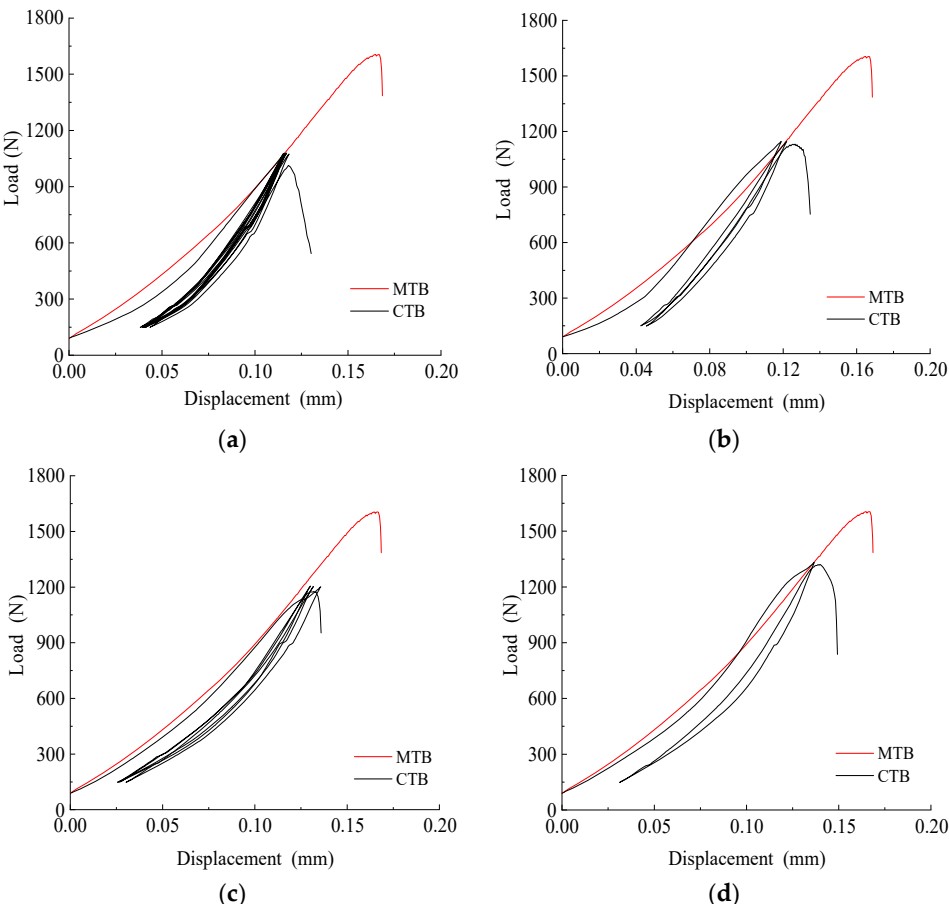

**Figure 5.** Load–displacement curves for CTB tests on granite at 400 °C under various upper load limits, with the MTB results provided as a reference. (**a**) 75%; (**b**) 80%; (**c**) 85%; (**d**) 90%.

**Table 3.** Mode I fracture toughness acquired during CTB tests.

| T/(°C) | MTB $K_{Ic}$/ (MPa·m$^{1/2}$) | ULL/(%) | Number of Cycles * | | | CTB $K_{Ic}$/ (MPa·m$^{1/2}$) | DA/(%) |
|---|---|---|---|---|---|---|---|
| | | | Sample 1 | Sample 2 | Sample 3 | | |
| 25 | 1.52 | 75 | 200 | 200 | 200 | - | - |
| | | 80 | 200 | 200 | 200 | - | - |
| | | 85 | 61 | 71 | | 1.24 | 18.25 |
| | | 90 | 11 | 36 | 13 | 1.28 | 14.19 |
| 200 | 1.27 | 75 | 200 | 200 | 200 | - | - |
| | | 80 | 200 | 200 | 200 | - | - |
| | | 85 | 27 | 16 | 12 | 1.06 | 16.29 |
| | | 90 | 8 | 23 | 15 | 1.06 | 16.23 |
| 300 | 1.11 | 75 | 200 | 200 | 200 | - | - |
| | | 80 | 29 | 30 | 9 | 0.80 | 27.81 |
| | | 85 | 5 | 49 | 136 | 0.91 | 19.68 |
| | | 90 | 33 | 15 | 6 | 0.98 | 14.46 |
| 400 | 1.05 | 75 | 8 | 11 | 17 | 0.77 | 27.14 |
| | | 80 | 18 | 3 | 4 | 0.85 | 19.38 |
| | | 85 | 13 | 12 | 6 | 0.84 | 19.55 |
| | | 90 | 4 | 3 | 4 | 0.82 | 13.36 |

Note: MTB $K_{Ic}$—average mode I fracture toughness from MTB tests; ULL—upper load limit; CTB $K_{Ic}$—average mode I fracture toughness from CTB tests. DA—decreasing amplitude (DA = $\frac{\text{MTB}K_{Ic} - \text{CTB}K_{Ic}}{\text{MTB}K_{Ic}} \times 100\%$). * The maximum cycle number is 200.

### 3.2. Cumulative Vertical Displacement

Figure 6 illustrates the curves correlating the CVD with the number of loading cycles at elevated temperatures. These CVD–number of loading cycle curves visually demonstrate the connection between the ultimate deformation of the granite and the number of loading cycles. The progression of CVD can be categorized into three distinct stages: an initial rapid growth stage, a subsequent stable growth stage, and finally, an accelerated growth stage. At 25 °C, the CVD of granite samples gradually increases as the number of loading cycles increases, as demonstrated in Figure 6. Notably, for samples exposed to an equivalent cycle number, the CVD exhibits an upward trend alongside an increase in the upper load limit. Furthermore, it is worth highlighting that failure and nonfailure samples exhibit differences in the stages of cyclic loading at which CVD occurs, as observed by Li et al. [46]. For instance, in cases where the samples do not fail, a significant portion of the CVD is concentrated in the initial phase of cyclic loading. In fact, the cumulative deformation resulting from the first 50 cycles accounts for approximately 60% of the total deformation.

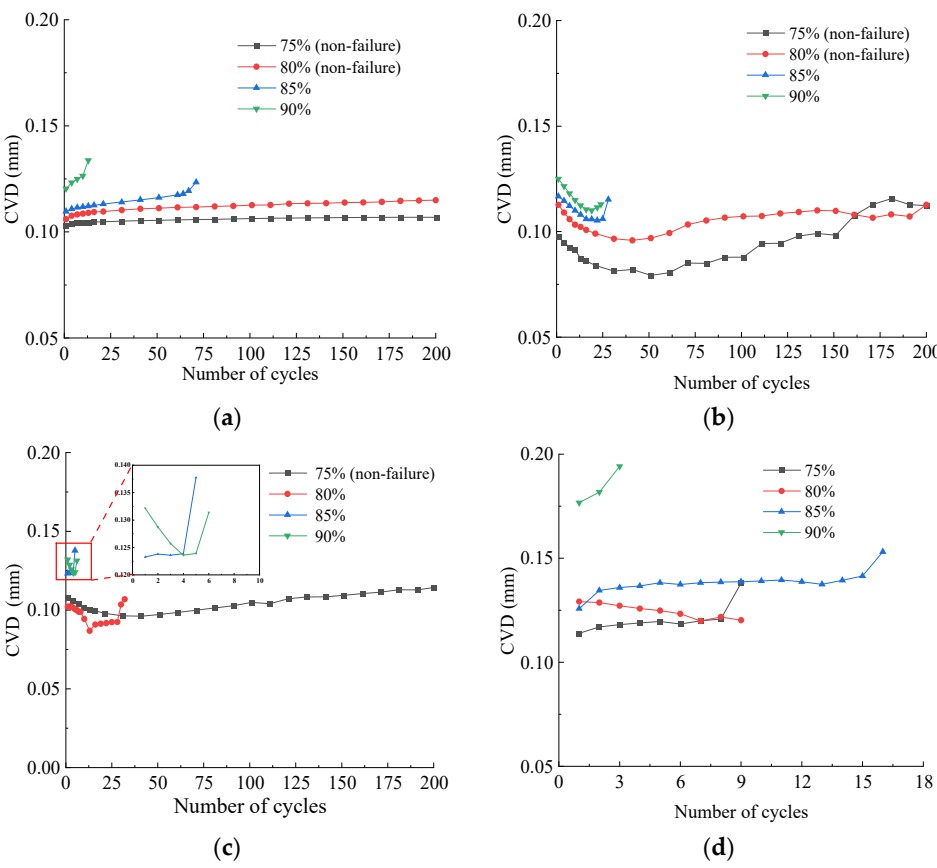

**Figure 6.** The cumulative vertical displacement (CVD) evolution curves with the number of loading cycles at different increased temperatures. (**a**) 25 °C; (**b**) 200 °C; (**c**) 300 °C; (**d**) 400 °C.

On the other hand, in cases where the samples do fail, the CVD primarily occurs during the failure stage, which corresponds to the latter part of the cyclic loading process and accounts for roughly 50% of the total displacement. Similar findings have also been reported in previous research conducted by Wang et al. [23] and Wichtmann et al. [47]. In the early loading phase, the internal primary fractures and cracks undergo continuous compaction, leading to irreversible plastic deformation. As a result, the curve exhibits a significant upward trend during this initial loading phase. In the failure stage, microcracks within the sample continue to propagate and merge over multiple cycles, while damage defects accumulate. In the few cycles preceding sample failure, a rapid escalation in CVD leads to a sudden sample failure when it reaches a critical value. It should be noted that

when the upper load limit is set to a low value, sample failure may not occur. Consequently, the CVD of the sample only displays the initial rapid growth and stable stages, with no occurrence of an accelerated growth stage.

In most cases, the CVD tends to rise with an increase in the number of loading cycles. Nevertheless, a different trend emerges at 200 °C and 300 °C, where the CVD of granite samples initially decreases before subsequently increasing as the number of loading cycles rises, as depicted in Figure 6b,c. These findings highlight temperature and cyclic loading impacts on granite's physical and mechanical characteristics. In the early phases of cyclic loading, the temperature elevation triggers sample expansion, resulting in the closure of internal microcracks and the enlargement of the compressive region, which is in agreement with the observations of Zhang et al. [48]. The compressive region expansion and the increased elastic modulus within the samples indicate a thermal hardening effect, as suggested by Ji et al. [49]. As the number of loading cycles increases, the sample's microcracks steadily increase, and the damage deformation gradually accumulates. Over time, the thermal expansion effect becomes insufficient to compensate for the accumulated plastic deformation, leading to a subsequent increase in CVD. As the sample approaches failure, a noticeable, sharp upturn in CVD is observed, as shown in Figure 6, which indicates sample failure. At 400 °C, the cumulative vertical deformation remains relatively stable during the initial cycles and significantly increases only upon sample failure. This behavior can be attributed to granite being in a brittle–ductile transition state at 400 °C, as Yin et al. [42] suggested.

In contrast to conditions at 200 °C and 300 °C, the plasticity of granite is enhanced at 400 °C. As a result, granite can undergo more substantial plastic deformation under a similar cyclic load. At 400 °C, this plastic deformation reaches an equilibrium state due to thermal hardening effects, resulting in a relatively steady CVD curve. This observation indicates that under cyclic loading and elevated temperatures, although the sample does not exhibit significant plastic deformation, internal fatigue damage remains an influential factor to consider.

### 3.3. Evolution of Elastic Stiffness

Elastic stiffness is a crucial mechanical property of rocks, often represented by the load–displacement curve's slope, which describes how rocks evolve at elevated temperatures. In this study, the elastic stiffness during the loading stage is estimated based on the slope of the secant, which covers approximately 50% to 75% of the peak load at the corresponding temperature, an approach applied in previous studies [43,50]. The variation in elastic stiffness of granite during the loading phase as a function of the number of cycles in the CTB tests is illustrated in Figure 7.

As the number of cycles increases, the elastic stiffness of the granite initially rises, followed by a stabilization phase and, ultimately, a decline. Notably, the elastic stiffness of specific nonfailure samples does not exhibit a significant decrease. Figure 7a further illustrates that the elastic stiffness of these nonfailure samples undergoes a significant enhancement as the number of cycles progresses. It escalates rapidly during the initial loading stage, reaching its maximum value after a few cycles. The initial cycles significantly impact the elastic stiffness, contributing to approximately 70–80% of the total increase. This is attributed to the substantial compaction of primary fractures and microcracks during these initial loading cycles, resulting in a notable improvement in the sample's compactness. Consequently, the elastic stiffness of the samples experiences a remarkable increase during the second loading cycle.

As the cyclic loading test progresses, the sample's internal structure undergoes restructuring, further enhancing the sample's compactness. During the intermediate phase of cyclic loading, the increase rate in elastic stiffness diminishes as the number of cycles rises. Ultimately, the elastic stiffness reaches a stable state with further increases in cycle numbers. This has been associated with increased friction between fissures, resulting from debris settling into the microcracks during loading, which induces stiffness augmentation [51,52].

Therefore, the initial loading cycles effectively boost the sample's elastic stiffness. However, as depicted in Figure 7, most samples have a subsequent reduction in elastic stiffness. During this phase, the internal microcracks and fatigue damage in the granite progressively accumulate with each cycle, resulting in increased plastic deformation and, consequently, a decline in elastic stiffness. Several studies have revealed that an augmentation in the upper load limit leads to a decline in the elastic stiffness of the tested specimens [31,53]. This occurrence can be ascribed to the acceleration of crack development and the consequent premature initiation of the elastic stiffness degradation phase caused by the heightened upper load limit. The findings of the current experiment effectively substantiate this claim. Additionally, with an increase in temperature, there is a noticeable downward shift in the elastic stiffness curves, as illustrated in Figure 7. This shift indicates a decrease in the elastic stiffness of granite as the temperature rises. This observation aligns with the experimental findings of Luo et al. [54]. Nevertheless, certain studies have discovered that the elastic stiffness of rocks exhibits an increase within the temperature range of 25–200 °C [55]. This finding further suggests the absence of a definitive pattern in the alteration of mechanical properties of rocks between 25 and 200 °C.

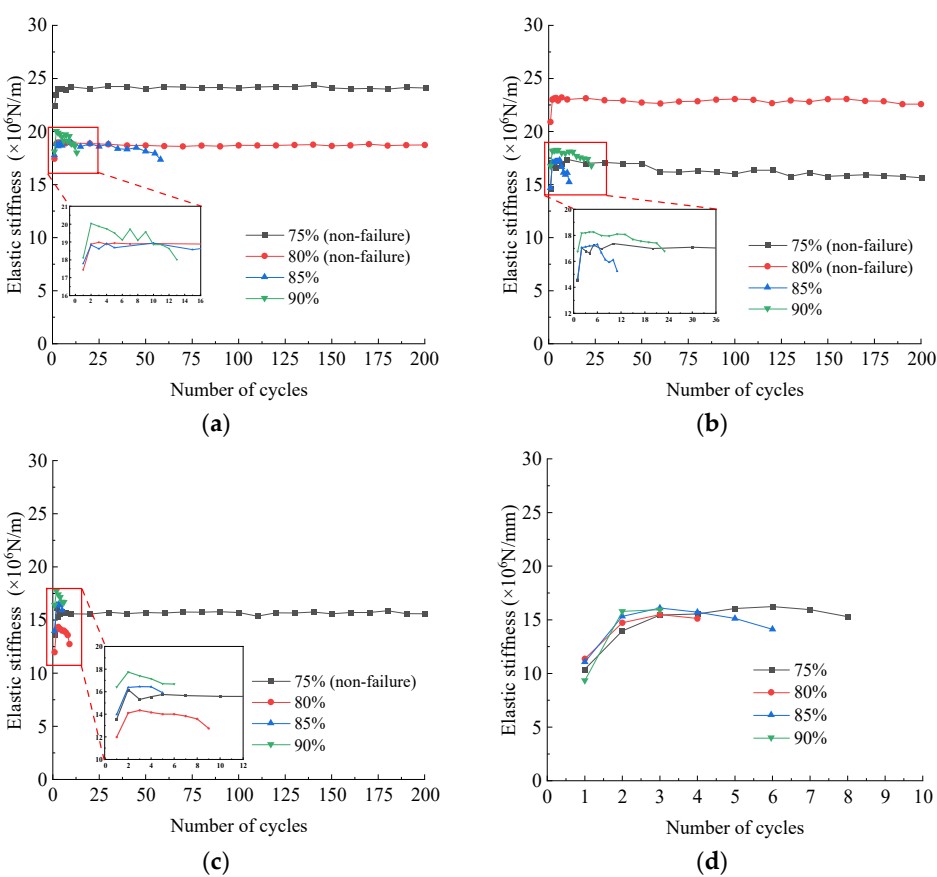

**Figure 7.** Evolution of granite's elastic stiffness with the number of cycles at different increased temperatures. (**a**) 25 °C; (**b**) 200 °C; (**c**) 300 °C; (**d**) 400 °C.

### 3.4. Mode I Fracture Toughness

Table 3 presents granite's mode I fracture toughness values as measured through MTB and CTB tests conducted under high-temperature conditions. These results indicated that cyclic loading has a pronounced and detrimental impact on granite's mode I fracture toughness. Notably, the maximum reduction in fracture toughness is substantial, reaching 18.25%, 16.29%, 27.81%, and 27.14% at the real-time temperatures of 25 °C, 200 °C, 300 °C, and 400 °C, respectively.

The fluctuations in mode I fracture toughness at different high temperatures concerning the upper load limit are depicted in Figure 8. Notably, the fracture toughness demonstrates an ascending pattern as the upper load limit increases at the same temperature. This can be attributed to the memory effect and the damage traits exhibited by the rock during cyclic loading. As such, recent research has confirmed that acoustic emission energy tends to be higher near the upper load limit during acoustic emission tests, as observed by Wang et al. [56]. This observation proves that significant fatigue damage occurs near the upper load limit. Consequently, the sample is more susceptible to failure near this limit. In this regard, Sample 3 failed after 136 cycles, a notable departure from the failure cycles of Samples 1 and 2, occurring at 300 °C and 85% of the load limit. Not only do such variations align with the randomness and heterogeneity of granite, but they are consistent with observations in engineering scenarios. The results presented above clearly indicate that the upper load limit has a particular effect on the fracture initiation pressure of the reservoir. Specifically, when the upper load limit surpasses the damage threshold, the initiation pressure increases with the increase of the upper load limit. Therefore, lower load limits can be considered for cyclic hydraulic fracturing, offering a potential opportunity to reduce the demands on mechanical equipment.

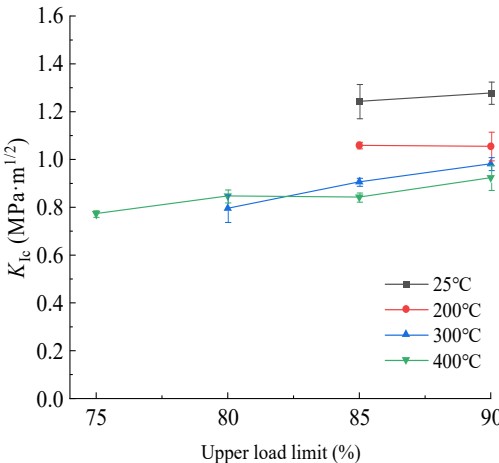

**Figure 8.** Evolution of mode I fracture toughness with upper load limit at different temperature settings.

In theory, an increase in the upper load limit of cyclic loading should result in elevated CVD and more significant accumulated plastic deformation in each cycle, ultimately leading to a gradual decrease in the number of cycles needed to cause sample failure. However, the observed reduction in the required number of cycles is not monotonically reduced in this study, as presented in Table 3. This variation may be attributed to the heterogeneity of the granite samples, causing differences between the theoretical upper load limit and the actual values in the cyclic load tests. Another contributing factor could be the relatively minor changes in the upper load limit across the four levels, which may have resulted in a lack of visible patterns between the number of cycles needed to induce sample failure and the upper load limit. Further in-depth investigations are recommended to gain a more comprehensive understanding of the influence of the upper load limit on the number of cycles required for failure.

The relationship between mode I fracture toughness and increasing temperature under different upper load limit conditions is visually presented in Figure 9. Notably, at 400 °C, all samples experience failure after cyclic loading. As the temperature decreases to 300 °C, samples under 80%, 85%, and 90% load limits fail, whereas in the range between 25 °C and 200 °C, only samples subjected to 85% and 90% load limits ultimately fail. These observations collectively indicate a reduction in mode I fracture toughness as the temperature increases.

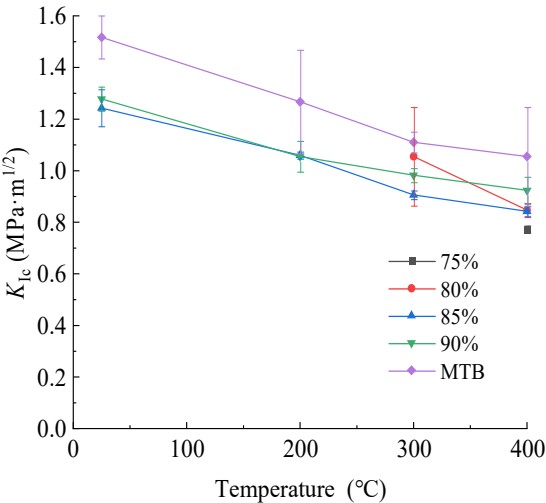

**Figure 9.** Evolution of mode I fracture toughness with increased temperatures under different upper load limits.

In order to characterize the fatigue damage of granite at various temperatures, the damage threshold is defined as the ratio of the fatigue damage load to the peak load of the MTB at the corresponding temperature. Comparing these findings to the work of Mo et al. [57], it becomes apparent that the upper load limit influences the damage threshold. The relationship between the damage threshold and temperature is graphically represented in Figure 10. Granite's damage threshold remains relatively stable within the temperature range of 25 °C to 200 °C. This phenomenon has been extensively observed in scholarly experiments [15,18–20]. The temperature range (25 °C to 200 °C) under consideration exhibits dual effects on the mechanical properties of granite, contributing to an overall tendency of stabilization (i.e., the thermal expansion effect enhances the mechanical properties, while the formation of thermal cracks and the loss of free water weaken the mechanical properties to some extent).

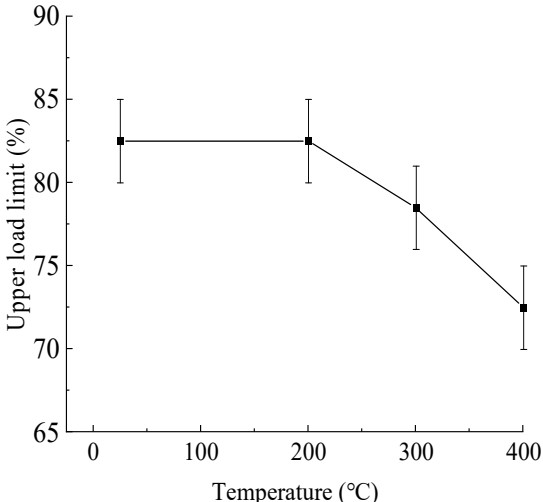

**Figure 10.** Variation of the damage threshold with increased temperature values.

However, as the temperature surpasses 200 °C, there is a notable decline in the damage threshold of granite with increasing temperature [58]. As the temperature escalates to 300 °C, the vaporization of bound water within the granite occurs. At this stage, the lubrication between internal particles of granite is diminished, causing an increase in friction between mineral particles during cyclic loading. Consequently, fractures occur after multiple cycles at a lower upper load threshold. The decline in granite's critical

damage threshold at this stage can be attributed to an increase in plasticity, leading to the effective accumulation of fatigue damage. Upon reaching 400 °C, the release of crystal water from the granite leads to the disruption of the internal microstructure of the granite, contributing to a noticeable reduction in the mechanical properties. The granite is weakened by the double effect of increased plasticity and decreased mechanical properties, leading to another decrease in the damage threshold.

Moreover, it has been proposed that the deterioration of granite can be attributed to the diminishing mechanical properties of its various components at varying temperatures and a weakening of the cementation between minerals [59,60]. The experimental results reveal a 30.89% reduction in the fracture toughness of granite at 400 °C compared to 25 °C in the MTB tests. Furthermore, in the CTB tests at 400 °C, the fracture toughness of granite decreases by 49.3% compared to the MTB tests at 25 °C. This reduction can be attributed to the combined impact of cyclic loading and high temperature, significantly reducing the fracture initiation pressure of HDR formations. Consequently, cyclic hydraulic fracturing of higher-temperature HDR reservoirs can substantially limit the initiation pressure and reservoir stress, ultimately reducing induced seismicity magnitude. Throughout the circulation process, the reservoir fractures undergo extensive development, resulting in the creation of a complex fracture network that enhances the efficiency of EGS when compared to monotonic injection methods, as discussed by Zang et al. [61] and Hofmann et al. [62].

*3.5. Fracture Surface Morphology*

During cyclic loading, the crack initially originates at the tip of the pre-existing crack and subsequently propagates along the loading direction toward the loading point.

As shown in Figure 11, at temperatures below 200 °C, the granite surface exhibits a gray-white color. When the temperature rises, the sample's color transitions gradually from gray-blue to gray-white, accompanied by yellow spots on the surface. This change is attributed to the transformation of black mica-rich granite to yellow at elevated temperatures, primarily due to dehydration [59,63]. The crack paths are detected using a camera photography system, as depicted in Figure 12. The crack deviation distance, which measures how far the crack path deviates from the pre-crack plane, is closely associated with the particle size and internal structure of the sample, as found by Kataoka et al. [64]. In the temperature range of 25 °C to 200 °C, the crack deviation distance is relatively small, and the crack path appears relatively straight, indicating apparent features of brittle fracture. However, as the temperature rises, the crack deviation distance and the tortuosity degree gradually increase. This is primarily due to the elevated temperature causing more significant thermal damage and the formation of transgranular cracks. The growing number of cracks and their unpredictable propagation direction contribute to increased crack path tortuosity. On the other hand, the decomposition of larger mineral particles significantly affects the crack path at elevated temperatures, as noted by Jiang et al. [65]. This analysis underscores that the impact of rising temperature on the crack path is more pronounced compared to cyclic loading.

Figure 13 shows the microcrack morphology of the granite after MTB and CTB under high temperature using SEM (500×), respectively. At 25 °C, as depicted in Figure 13a, a few secondary pores and primary cracks are visible on the fracture plane of the granite sample under MTB, indicating that the granite structure remains nearly intact and compact. It should be noted here that most main cracks are intercrystalline. This phenomenon aligns with the findings of Song et al. [66], who also observed numerous micropores on the failure surface following cyclic loading, a characteristic not detected in the present test. As the temperature reaches 200 °C, intergranular cracks develop on the fractured surface, which can be attributed to the differential expansion between various mineral grains at high temperatures, leading to higher stress concentration at the crack tip. Eventually, the concentrated stress causes the breakage of the mineral grains at the end of the crack, inducing an increase in the length and width of the intergranular crack, which is consistent with the findings of Yang et al. [67]. Within the temperature range of 25 °C to 200 °C, the

fracture morphology of the specimen exhibits overall integrity, with intercrystalline cracks predominantly present, indicative of a characteristic brittle fracture [22].

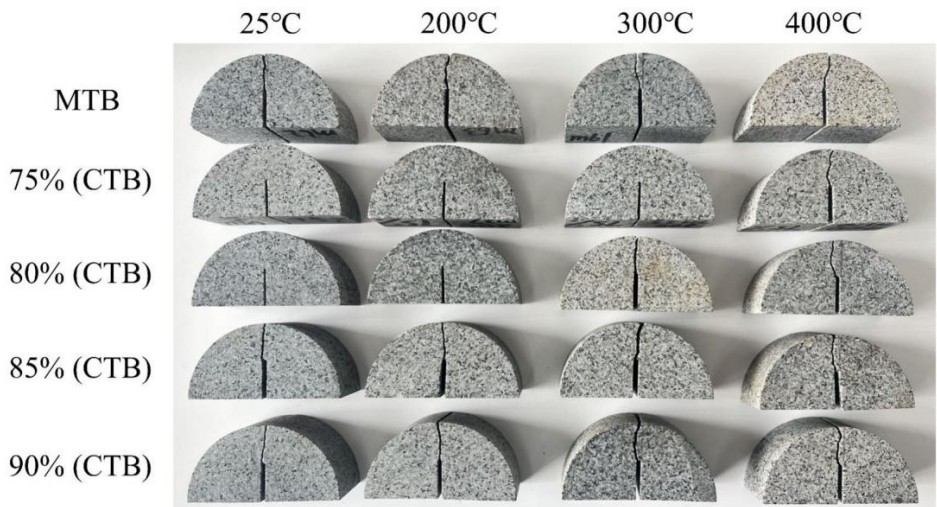

**Figure 11.** Photos of SCB samples under different upper load limits and increased temperature settings.

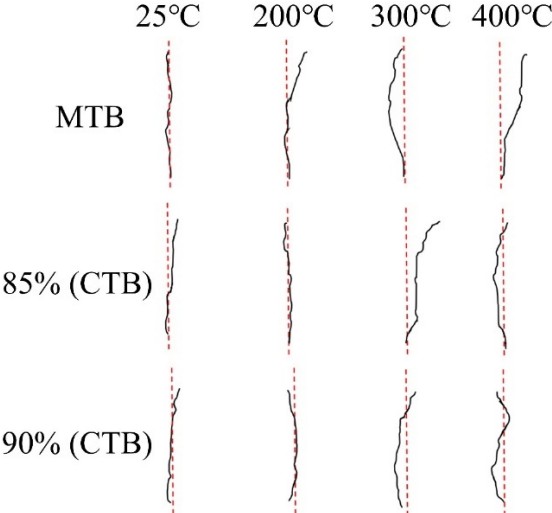

**Figure 12.** Crack path of SCB samples under 85% and 90% upper load limits.

At 300 °C and 400 °C, the differential expansion between mineral grains becomes more significant, and the prevalence of intergranular cracks increases considerably. As intergranular cracks develop, they tend to intersect and interconnect, leading to the loosening of mineral grains at these crack intersections [42,68]. This process contributes to the structural degradation of the granite sample, where mineral grains are exfoliated, and larger pores and crystal fracture zones are formed. SEM imaging results reveal the presence of numerous microcracks and crystal fracture zones resulting from exposure to high temperatures, leading to the degradation of the rock's microstructure, which degrades the macro-mechanical properties of granite.

During cyclic loading, the dislocation and shearing between mineral grains within the rock intensify, leading to the rapid development of the original secondary cracks, which eventually merge with the main crack [69,70]. As a result, there is a notable increase in the number and depth of cracks on the fracture surface compared to those in the MTB tests. Meanwhile, the alternating stress from cyclic loading intensifies the extrusion and friction between mineral grains on either side of the intergranular crack [71]. This effect

separates certain mineral grains from the rock matrix, causing grain decohesion and their scattering on the fracture surface, as illustrated in Figure 13b,d. With an increase in the number of loading cycles, the number of scattered grains and crystal fracture zones also increases, triggering significant changes in the topography of the granite fracture surface. This research demonstrates that granite failure under cyclic loading occurs due to the coalescence of numerous microcracks and grain decohesion rather than the growth of a single macrocrack. Based on the fracture morphology, the presence of numerous microcracks and crystal fracture zones suggests a transition towards ductile fracture. However, the mechanical test results still demonstrate a predominantly abrupt fracture trend, indicating that the granite is currently undergoing a brittle-to-ductile transformation stage [22,72]. Yang et al.'s experimental results demonstrated that as the temperature increases up to 500 °C, the fracture behavior of granite transitions towards ductile fracture [73].

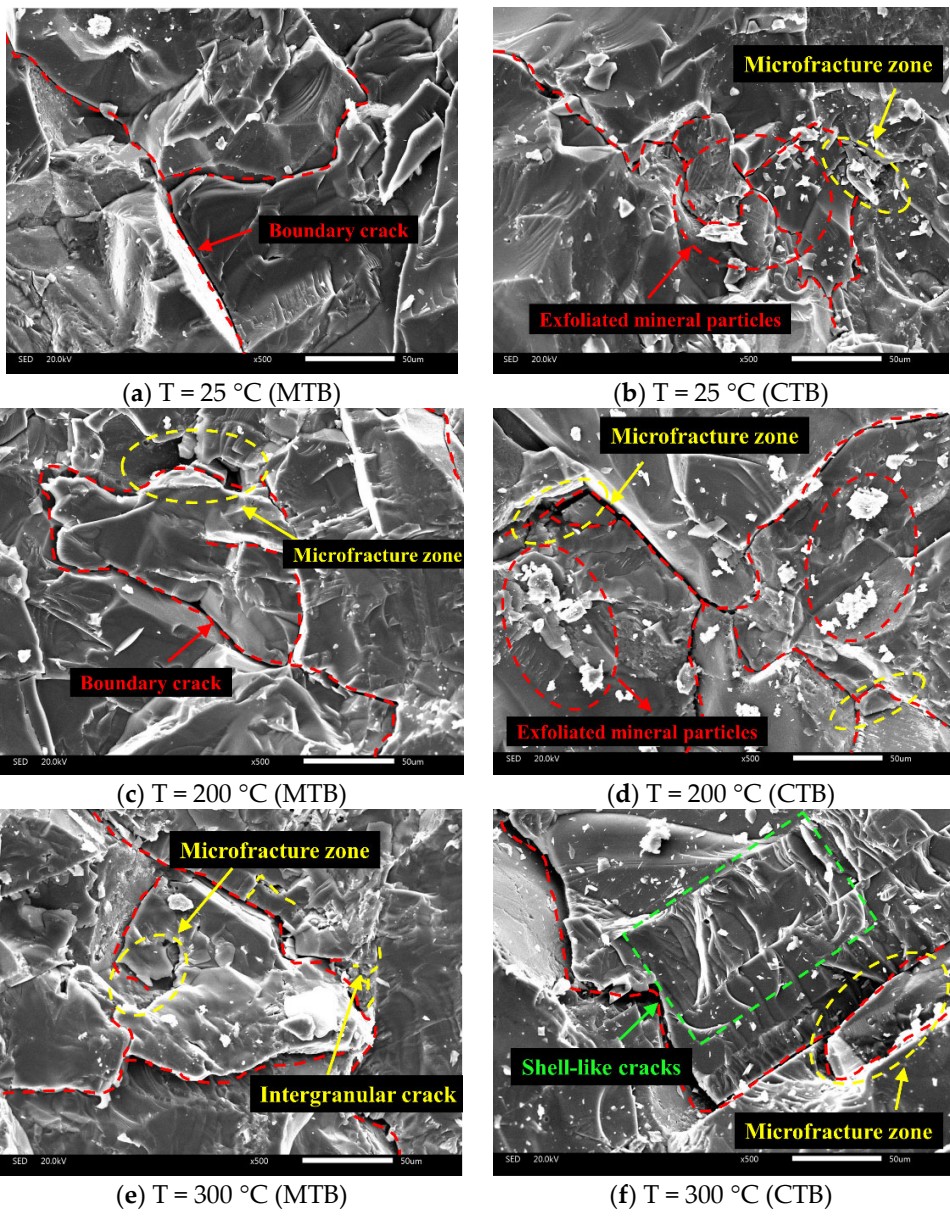

(**a**) T = 25 °C (MTB)

(**b**) T = 25 °C (CTB)

(**c**) T = 200 °C (MTB)

(**d**) T = 200 °C (CTB)

(**e**) T = 300 °C (MTB)

(**f**) T = 300 °C (CTB)

**Figure 13.** *Cont.*

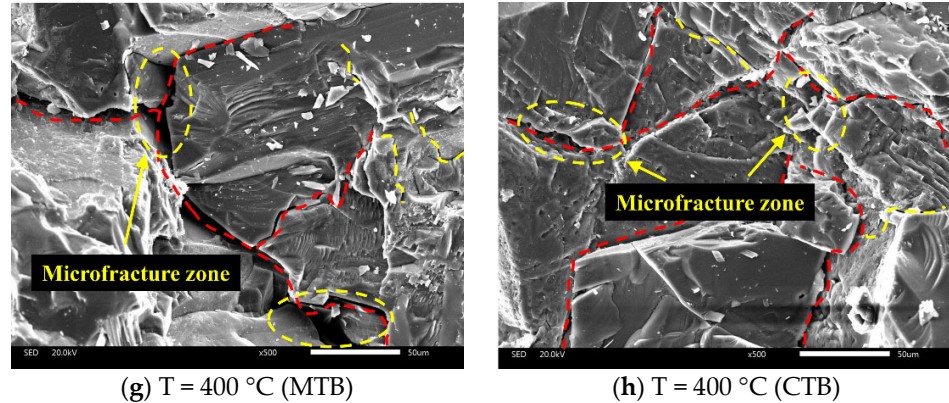

(**g**) T = 400 °C (MTB)          (**h**) T = 400 °C (CTB)

**Figure 13.** SEM images of the SCB fracture surface at varying high temperatures: (**a**,**c**,**e**,**g**) depict MTB samples, whereas (**b**,**d**,**f**,**h**) depict CTB samples (85% upper load limit). In these images, red dashed lines delineate grain boundary cracks, yellow dashed lines represent intergranular cracks, red circles highlight exfoliated mineral particles on the crystal surface, yellow circles indicate microfracture zones, and green rectangles outline shell-like cracks.

Interestingly, when the upper load limit is set at 85% at 400 °C, the significant reduction in the required number of cycles results in fewer exfoliated particles, ultimately leading to minimal disparity in the microstructures of the fracture surfaces after MTB and CTB tests. It should be noted that during cyclic loading, the heterogeneity of the granite microstructure causes a localized deformation and slip surfaces, introducing new shell-like cracks similar to those observed by Khvorstyanyi et al. [74] and Huang et al. [75]. At the microscopic level, these findings demonstrate the significant impact of high temperature and cyclic loading on the damage to the granite microstructure.

## 4. Conclusions

This research investigated the effect of cyclic loading on the load–displacement curves, elastic stiffness, the mode I fracture toughness and the crack development of granite at different real-time elevated temperatures (i.e., 25 °C, 200 °C, 300 °C, and 400 °C) through three-point bending tests. Several vital conclusions are presented as follows:

1. At a temperature of 25 °C, the CVD of the granite progressively increased with an increase in the number of cycles, primarily accumulating during the early and damage stages of cyclic loading. In the temperature range of 200 °C to 400 °C, the thermal hardening effect led to a trend in the CVD of granite, shifting from a decrease to an increase with an increasing number of cycles.
2. Under cyclic loading, the elastic stiffness of granite exhibited a pattern of increasing–stabilizing–decreasing with a growing number of cycles. After the initial loading cycle, granite stiffness increased substantially by 70% to 80%.
3. When temperatures surpassed 200 °C, two substantial reductions were observed in the damage threshold. At 300 °C, the threshold dropped below 80% of the upper load limit, and it further decreased to less than 75% as the temperature reached 400 °C.
4. The microstructure of granite was significantly influenced by high temperature and cyclic loading. Below 300 °C, a comparison of SEM images from fracture surfaces in MTB and CTB tests revealed an increase in microfracture zones, cracks, scattered particles, and the size of scattered particles after cyclic loading.
5. Based on the analyses above, this study can serve as a valuable reference for cyclic hydraulic fracturing technology. For instance, a lower upper limit of circulating load can be chosen within a reasonable range to minimize the impact of mining on the geothermal reservoir, thereby reducing the risk of induced earthquakes. Alternatively, hydraulic fracturing can be employed to extract HDRs at the optimal temperature, creating a complex fracture network and thereby enhancing extraction efficiency. In

future work, it is recommended to examine the effect of real-time elevated temperatures on the damage threshold, potentially considering even higher temperatures. Given that the maximum number of cycles was limited in the present study, further research will be conducted to explore experiments with an increasing number of cycle loads.

**Author Contributions:** Conceptualization, F.L. and F.Z.; formal analysis, F.L., S.Z. and K.L.; data curation, F.L. and S.M.; writing—original draft preparation, F.L.; writing—review and editing, F.L. and F.Z.; supervision, F.Z. The results were discussed and conclusions were drafted jointly by all authors. All authors have read and agreed to the published version of the manuscript.

**Funding:** This work was jointly supported by the National Natural Science Foundation of China (Grant No. 51979100) and Shandong Lunan Geological Engineering Exploration Institute Open Fund (Grant No. LNY2020-Z08).

**Institutional Review Board Statement:** Not applicable.

**Informed Consent Statement:** Not applicable.

**Data Availability Statement:** The raw data supporting the conclusions of this article will be made available by the authors on request.

**Acknowledgments:** The authors are indebted to Yinlin Ji of Helmholtz Centre Potsdam GFZ German Research Centre for Geosciences for their great help in review.

**Conflicts of Interest:** The authors declare no conflict of interest.

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
