# Peer review of "Effect of Cyclic Loading on Mode I Fracture Toughness of Granite under Real-Time High-Temperature Conditions"

_applsci, doi:10.3390/app14020755_

Round 1

Reviewer 1 Report

Comments and Suggestions for Authors

Opinion

The article has relevance and merit because it deals with fracture toughness in granitic rocks subjected to high temperatures and cyclic stresses. For the work to be published, some changes need to be made.

1) The abstract meets the objectives of the article.

2) The introduction is well-written with up-to-date bibliographical references;

3) Methodology meets the objectives of the article.

4) Figure 7a, b c and d - The results need a better discussion or theoretical basis in relation to the literature of the last 5 years.   

5) Figure 13 (SEM)- Needs better discussion about microstructure and comparison with literature data from the last 5 years.

6) Conclusion- needs to be modified to meet the objective of the work.

7) This reviewer believes that the work should be published after modifications.

Reviewer 2 Report

Comments and Suggestions for Authors

This manuscript deals with  the effect of cyclic loading on mode I fracture toughness of granite under real-time high-temperature conditions. It is an interesting study suitable for publication in this journal. It is well organized and the results well presented. Nevertheless more mineralogical data from the initial samples as well as after their temperature treatmment could be  included. Please add such data, if you have,  or refer if you have plans to present in a future study. In my opinion such data will enrich your results and their  interepretation as well as the discussion.

 Please add in Fig 13, if it is  a BSE or SEI photo?

Kind regards

Reviewer 3 Report

Comments and Suggestions for Authors

The manuscript presents an experimental study on the effect of cyclic loading on the fracture toughness of rock under various temperatures. The manuscript could be considered for publication in this journal after addressing the following comments:

1. The novelty of this experimental study should be clearly demonstrated in the introduction section.

2. In the section of Materials and Methods, the information about the testing method for SEM, including equipment and parameters, is missing. 

3. On page 18, how did the authors differentiate the microcracks caused by high temperature or cracks propagation during the cyclic loading from the SEM image?

4. I did not find any papers in the References from the Applied Sciences. Since the authors submitted the manuscript to this journal, I recommend that the authors should incorporate some papers from Applied Sciences into this study.  

Reviewer 4 Report

Comments and Suggestions for Authors

The study investigates the response of granite to cyclic loading at temperatures ranging from 25°C to 400°C. Three-point bending tests reveal an increase in cumulative vertical displacement with cycles, a declining critical damage threshold with rising temperatures, and notable microscopic changes. The findings provide insights into granite behavior under extreme conditions, particularly relevant to cyclic hydraulic fracturing technology in hot dry rock development. The paper is interesting and can be published if the authors address the following comments:

Mechanisms and Phenomena: Can you elaborate on the mechanisms or phenomena that might contribute to the observed decline in the critical damage threshold of granite as the temperature increases from 25°C to 400°C? Were there any unexpected behaviors or anomalies in the load-displacement curves, elastic stiffness, or fracture toughness at extreme temperatures (e.g., 400°C)?

Microscopic Features: Can you discuss any correlations between the observed microscopic features (e.g., cracks, crystal microfracture zones, dislodging of mineral particles) and the cyclic loading conditions at different temperatures?

Trends and Patterns: Were there any clear trends or patterns in the load-displacement curves or fracture toughness that can be attributed specifically to the number of cycles, especially within the temperature range of 25°C to 300°C?

Implications for Hydraulic Fracturing: How do the observed behaviors of granite under cyclic loading and high-temperature conditions contribute to the understanding of challenges faced in cyclic hydraulic fracturing in hot dry rock development? What practical implications or insights do these findings provide for the optimization of cyclic hydraulic fracturing technology?

Figure 7: Please explain this figure in more detail and compare it with related literature in this area and published papers.

Table 3: Please fix the positioning; it's currently not within the frame.

Test Conditions: What is the room temperature in your tests? Were all the tests conducted under the same conditions?

Comments on the Quality of English Language

The study investigates the response of granite to cyclic loading at temperatures ranging from 25°C to 400°C. Three-point bending tests reveal an increase in cumulative vertical displacement with cycles, a declining critical damage threshold with rising temperatures, and notable microscopic changes. The findings provide insights into granite behavior under extreme conditions, particularly relevant to cyclic hydraulic fracturing technology in hot dry rock development. The paper is interesting and can be published if the authors address the following comments:

Mechanisms and Phenomena: Can you elaborate on the mechanisms or phenomena that might contribute to the observed decline in the critical damage threshold of granite as the temperature increases from 25°C to 400°C? Were there any unexpected behaviors or anomalies in the load-displacement curves, elastic stiffness, or fracture toughness at extreme temperatures (e.g., 400°C)?

Microscopic Features: Can you discuss any correlations between the observed microscopic features (e.g., cracks, crystal microfracture zones, dislodging of mineral particles) and the cyclic loading conditions at different temperatures?

Trends and Patterns: Were there any clear trends or patterns in the load-displacement curves or fracture toughness that can be attributed specifically to the number of cycles, especially within the temperature range of 25°C to 300°C?

Implications for Hydraulic Fracturing: How do the observed behaviors of granite under cyclic loading and high-temperature conditions contribute to the understanding of challenges faced in cyclic hydraulic fracturing in hot dry rock development? What practical implications or insights do these findings provide for the optimization of cyclic hydraulic fracturing technology?

Figure 7: Please explain this figure in more detail and compare it with related literature in this area and published papers.

Table 3: Please fix the positioning; it's currently not within the frame.

Test Conditions: What is the room temperature in your tests? Were all the tests conducted under the same conditions?
